# UV Irradiation-Induced SERS Enhancement in Randomly Distributed Au Nanostructures

**DOI:** 10.3390/s20143842

**Published:** 2020-07-09

**Authors:** Dong-Jin Lee, Dae Yu Kim

**Affiliations:** 1Inha Research Institute for Aerospace Medicine, Inha University, Incheon 22212, Korea; voinaimir82@gmail.com; 2Department of Electrical Engineering, College of Engineering, Inha University, Incheon 22212, Korea

**Keywords:** surface-enhanced Raman scattering, UV treatment, negatively charged adsorbed oxygen species, analyte–surface affinity, Au nanostructures

## Abstract

Currently used platforms for surface-enhanced Raman scattering (SERS) sensors generally employ metallic nanostructures for enrichment of the plasmonic hotspots in order to provide higher Raman signals, but this procedure is still considered challenging for analyte–surface affinity. This study reports a UV irradiation-induced SERS enhancement that amplifies the interactions between the analytes and metallic surfaces. The UV light can play critical roles in the surface cleaning to improve the SERS signal by removing the impurities from the surfaces and the formation of the negatively charged adsorbed oxygen species on the Au surfaces to enhance the analyte–surface affinity. To evaluate this scenario, we prepared randomly distributed Au nanostructures via thermal annealing with a sputtered Au thin film. The UV light of central wavelength 254 nm was then irradiated on the Au nanostructures for 60 min. The SERS efficiency of the Au nanostructures was subsequently evaluated using rhodamine 6G molecules as the representative Raman probe material. The Raman signal of the Au nanostructures after UV treatment was enhanced by up to approximately 68.7% compared to that of those that did not receive the UV treatment. We expect that the proposed method has the potential to be applied to SERS enhancement with various plasmonic platforms.

## 1. Introduction

Boosting the Raman signals in surface-enhanced Raman scattering (SERS) sensors has been extensively studied for decades, and the SERS sensors typically employ metallic nanostructures that generate plasmonic hotspots to concentrate the localized electromagnetic fields in the vicinity of the plasmonic nanostructures [1,2,3,4,5]. Various metallic nanostructures have been developed thus far for amplifying the plasmonic hotspots, including low-dimensional metallic nanostructures with highly localized electromagnetic field and two-/three-dimensional architectures with increased surface areas [5,6,7,8,9,10,11]. In addition, new analytical tools, including tip-enhanced Raman spectroscopy and shell-isolated nanoparticle-enhanced Raman spectroscopy, have been proposed for single-molecule detection with spatial resolution on the order of the sub-diffraction limit [5,12,13,14,15,16].

Despite extensive studies on hotspot engineering, a majority of the current SERS platforms rarely consider the effects of analyte adsorption on the metallic surfaces [2,17,18]. Poor analyte affinities to the metallic surfaces are quite problematic, because a high-performance SERS platform can be useless for detecting the analytes that do not interact with the electromagnetic hotspots. This remains a formidable issue for detecting analytes that have no interactions with metallic surfaces [2]. Therefore, strategies for enhancing the analyte–surface affinities are required for greater adoption of SERS sensors for practical applications. So far, four main methodologies have been presented to enhance the analyte affinities to the metallic surfaces, including chemical interactions between the analytes and the metallic surfaces, physical confinement of the analytes close to the metallic surfaces, encapsulation of the metallic surfaces for the concentration of the analytes on the electromagnetic hotspots, and chemical modifications of the analyte molecules [2]. Among them, the surface charges generated on the plasmonic surfaces can play a critical role for enhancing the analyte–surface affinities to promote the adsorption of polar as well as non-polar molecules [19,20,21].

UV–ozone treatment is an effective, simple, and powerful tool for surface modification and activation, which generates the reactive oxygen species on the film surfaces [22,23,24,25,26]. UV–ozone treatment cleans the metallic surfaces to improve the SERS signal by removing the impurities on the surfaces, facilitating the adsorption of analyte molecules [27,28,29]. Furthermore, UV–ozone treatment can be expected to induce the negatively charged adsorbed oxygen species on the Au surfaces. Saliba et al. reported that the electrons transfer from the Au surface into the adsorbed oxygen layer [30], and Sun et al. theoretically studied the relation between the adsorbed oxygen atoms and the Au surface atoms using the Hückel theory and density functional theory (DFT) calculations [31]. In addition, Hai et al. reported the formation of the surface charges on the Au surface due to the oxygen adsorption [20], and these negatively charged adsorbed oxygen species could play a role in the enhancement of the analyte–surface affinities [21]. Therefore, UV–ozone exposure can be used to boost the SERS spectra by the UV–ozone cleaning of the surface and the UV–ozone induced negatively charged adsorbed oxygen species.

In this work, we present an alternative strategy for SERS enhancement utilizing UV irradiation-induced surface cleaning and negatively charged adsorbed oxygen species for amplifying the interactions between the analytes and metallic surfaces. To this end, we prepared randomly distributed Au nanostructures via thermal annealing with a sputtered Au thin film. The UV light of central wavelength 254 nm was irradiated on the Au nanostructures for 60 min to enhance the analyte–surface affinity. The SERS enhancement was measured using rhodamine 6G (R6G) molecules as the representative Raman probe material, and a Raman signal increase of up to approximately 68.7% was observed. The proposed method therefore has the potential to be applied to SERS enhancement on various plasmonic platforms.

## 2. Experimental Procedures

### 2.1. Materials

Silicon wafers were purchased from Silicon Materials Inc., USA (B-doped p-type wafer, resistivity of 1–30 Ω∙cm). Rhodamine 6G (R6G) was purchased from Sigma-Aldrich Inc., South Korea; the Au target was purchased from Thifine Inc., Incheon, South Korea (2-inch diameter, purity of 99.99%).

### 2.2. Fabrication

The silicon wafers were cleaned using acetone, isopropyl alcohol, and deionized water in an ultrasonication bath. Then, Au thin films of thickness 3.4, 4.5, and 6.4 nm were deposited on the Si wafers via direct current (DC) sputtering with a base pressure of 5 × 10^−6^ Torr. A constant DC voltage of 350 V was applied, and argon was used to produce the plasma with 10 sccm. To fabricate the randomly distributed Au nanostructures, the Au thin film was thermally annealed on a hotplate at temperatures of 150, 250, and 350 °C under ambient conditions for 60 min. The samples were cooled freely in ambient conditions, and a UV lamp (2 × 15 W, UVITEC, Cambridge, UK) generating light with a central wavelength of 254 nm was used to irradiate the randomly distributed Au nanostructures for 60 min. After UV treatment, the Au nanostructures were soaked in rhodamine 6G aqueous solution of concentration 1 M for 60 min.

### 2.3. Characterization and Measurements

The morphologies of the randomly distributed Au nanostructures were characterized using scanning electron microscopy (SEM; Hitachi S-4300SE, Hitachi, Tokyo, Japan) and atomic force microscopy (AFM; NanoScope IV, Bruker, MA, USA). Raman spectra were measured via Raman spectroscopy (LabRAM HR Evolution, HORIBA, Kyoto, Japan) using an excitation wavelength of 532 nm, a laser power of 2 mW, an acquisition time of 5 s, and an accumulation of 1. The Raman spectra were obtained in the range of 400–1800 cm^−1^ for all experiments.

## 3. Results and Discussion

Figure 1a presents a schematic of the fabrication process for the randomly distributed Au nanostructures. The Au thin film was deposited via DC sputtering by adjusting the deposition time to control Au film thickness. The Au thin film was then aggregated into different sizes and shapes through thermal annealing. The UV light with a central wavelength of 254 nm was irradiated on the Au thin film for 60 min to fabricate the randomly distributed Au nanostructures. The SERS performances of the Au nanostructures irradiated with or without UV light were evaluated using R6G molecules adsorbed on the Au nanostructures, as shown in Figure 1b.

Figure 2 shows the SEM and AFM images of the randomly distributed Au nanostructures depicting the effect of initial thickness (3.4, 4.5, and 6.4 nm) and annealing temperature (150, 250, and 350 °C). The film thickness was determined from the cross-sectional profile of the film on a Si substrate after scratching a section of Au thin film (see Figure A1 in Section A.1). Uniformly distributed Au nanograins were observed for samples of Au nanostructures annealed at room temperature (RT). The root mean-squared (RMS) roughness value for Au3.4, Au4.5, and Au6.4 samples at RT were 0.1835 nm, 0.2264 nm, and 0.25 nm, respectively. With an increase in the annealing temperature, the grain sizes of the Au nanostructures increased. For the Au3.4 sample, the RMS roughness increased from 0.1835 nm at RT annealing to 0.43 nm at 350 °C annealing. For the Au4.5 sample, the RMS roughness increased from 0.2264 nm at RT annealing to 0.9979 nm at 350 °C annealing. For the Au6.4 sample, the RMS roughness increased from 0.25 nm at RT annealing to 9.329 nm at 350 °C annealing. In addition, it was observed that in the case of Au nanostructures having large initial thickness (Au6.4), the grain size and RMS roughness rapidly increased (for more detailed information, see Figure A2 in Section A.2).

Prior to evaluation of the SERS performances of the Au nanostructure by UV treatment, the Raman spectra depending on annealing temperatures and initial thicknesses were investigated in order to elucidate the SERS effect for these Au nanostructures. Figure 3a shows the Raman spectrum of the Au4.5 sample annealed at RT. This spectrum shows strong peaks of the vibrational bands at approximately 611, 773, 1310, 1363, and 1651 cm^−1^, corresponding to the Raman characteristic peaks of R6G [32,33,34]. The vibrational band at 611 cm^−1^ is due to the in-plane and out-of-plane xanthene ring deformations. The vibrational band at 773 cm^−1^ is due to the out-of-plane C–H bending and in-plane xanthene ring deformations. The vibrational band at 1310 cm^−1^ is due to the in-plane xanthene ring breath and in-plane N–H bending. The vibrational bands at 1363 and 1651 cm^−1^ originated from the xanthene ring stretching and in-plane C–H bending. The Raman peak of around 611 cm^−1^ is the most dominant R6G characteristic peak, and we selected this peak for comparison of the SERS activities of the randomly distributed Au nanostructures. Figure 3b shows the Raman signals of the Au3.4, Au4.5, and Au6.4 samples at 611 cm^−1^, depending on the annealing temperatures. A film formed by the evaporation process grows according to the following four stages: nucleation, growth of the nuclei and formation of larger islands, coalescence of the islands and formation of a connected network containing empty channels, and filling of the channels [35]. According to the aforementioned film-growth evolution, it can be expected that the plasmonic hotspots may diminish as the Au initial film thickness increases, thereby causing the SERS performance to deteriorate. For RT annealing, the Raman signals of the Au3.4 and Au4.5 samples are approximately twice that of the Au6.4 sample, presumably owing to the decrease in plasmonic hotspots. With the increase in annealing temperature, the Raman signals show a decreasing trend for the Au3.4, Au4.5, and Au6.4 samples, owing to gap widening between the Au nanostructures, thus diminishing the plasmonic hotspots.

To investigate the SERS performances of the Au nanostructures after UV treatment, the Raman signals from 10 samples of the Au nanostructures were compared as a function of UV treatment, as depicted in Figure 4. The Raman signals generally increased in the UV-treated Au nanostructures compared to the UV-untreated Au nanostructures. The SERS enhancement in each case was calculated as $\left(I_{UV}-I_{0}\right)/I_{0}\;(\%)$, where $I_{UV}$ and $I_{0}$ are the Raman intensities with and without UV treatment, respectively. Table 1 shows the Raman signal enhancement of Au nanostructures before and after UV treatment. The Raman signal of the Au4.5 sample annealed at 350 °C increased to about 68.75% after UV treatment, showing the greatest Raman signal enhancement. The SEM and AFM images were also compared to investigate the morphological variations of the Au nanostructures from UV treatment. As illustrated in Figure A3 in Section A.3, the morphological changes of the Au nanostructures with and without UV irradiation were not readily revealed. The RMS roughness values of the Au4.5 sample with and without UV treatment were about 1.745 nm and 1.776 nm, respectively.

UV–ozone treatment is expected to function as surface cleaning for the removal of the impurities on the metallic surfaces and the formation of the negatively charged adsorbed oxygen species. UV-induced surface cleaning is a well-known process, and plays a certain role of dissociating directly the C–C and the C–H bonds in the adsorbed organic molecules on the Au surfaces, as detailed in the literature [36,37,38]. This cleaning effect enables more adsorption of R6G molecules onto Au nanostructures and may contribute to the improvement of the SERS signal. In addition, the UV-induced negatively charged adsorbed oxygen species can also cause an enhancement in the SERS signal. Figure 5 shows the schematic of the mechanism of the SERS enhancement by the UV irradiation-induced negatively charged adsorbed oxygen species, thereby enabling the analyte–surface affinity. Under UV irradiation of the Au nanostructures, the excited electrons generated from the Au bulk are transferred into the adsorbed oxygen ions, generating the negatively charged adsorbed oxygen species on the surfaces of the Au nanostructures. The negatively charged Au nanostructures seem to undergo an attractive electrostatic interaction with the R6G molecules. This interaction enhances the adsorption of the R6G molecules, resulting in the enhancement of the SERS intensities.

## 4. Conclusions

We proposed and demonstrated a new strategy for SERS enhancement utilizing UV treatment for amplifying the interactions between analytes and metallic surfaces. The UV irradiation seems to function as surface cleaning and promote the formation of the negatively charged adsorbed oxygen species on the surfaces of Au nanostructures. These effects are expected to enhance the analyte–surface affinity and enable the SERS enhancement, but it has not been clarified yet which one is more dominant. The SERS performance using this proposed method was measured using R6G molecules, and a Raman signal increase of up to approximately 68.7% was observed. The proposed method therefore has the potential to be applied to SERS enhancement on various plasmonic platforms.

## Figures and Tables

**Figure 1 sensors-20-03842-f001:**
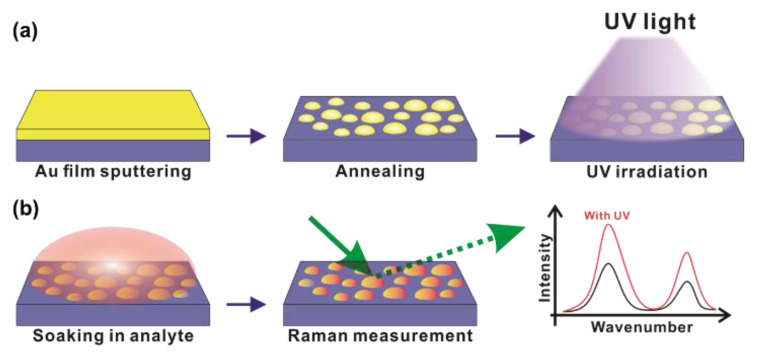
Schematic of UV irradiation-induced surface-enhanced Raman scattering (SERS) enhancement: (**a**) fabrication of the randomly distributed Au nanostructures by thin film sputtering, thermal annealing, and UV treatments; (**b**) SERS performance of the Au nanostructures irradiated with or without UV light and evaluated using rhodamine 6G (R6G) molecules as the Raman probe material.

**Figure 2 sensors-20-03842-f002:**
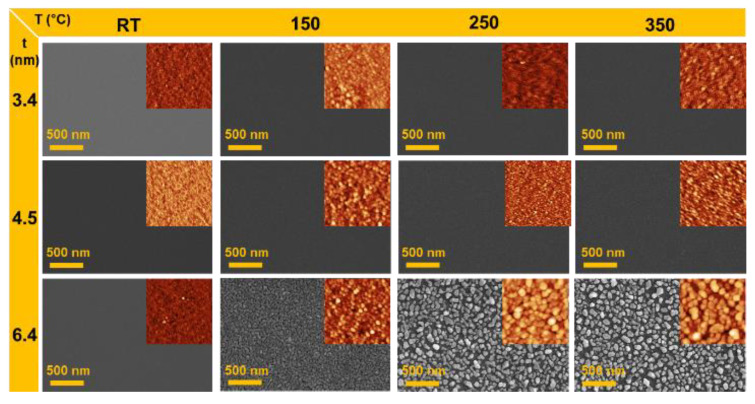
SEM and atomic force microscopy (AFM) images of randomly distributed Au nanostructures, showing the grain size variation of Au nanostructures with annealing temperature and initial Au film thickness. Insets: The size of AFM images are all 1 × 1 µm^2^.

**Figure 3 sensors-20-03842-f003:**
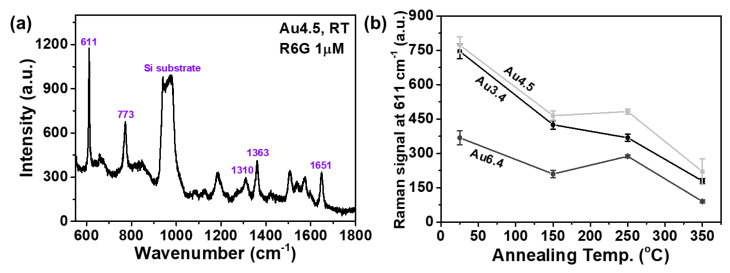
Raman spectra for 1 µM of R6G adsorbed on the randomly distributed Au nanostructures. (**a**) Raman spectrum of the Au4.5 sample annealed at RT. (**b**) Raman signals of the Au3.4, Au4.5, and Au6.4 samples at 611 cm^−1^ as a function of the annealing temperatures.

**Figure 4 sensors-20-03842-f004:**
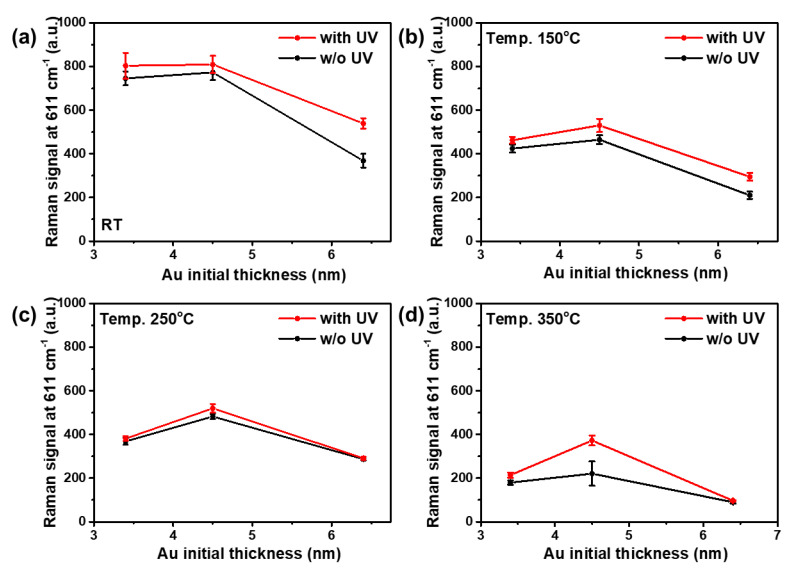
Raman signals of the Au3.4, Au4.5, and Au6.4 samples with and without UV treatment for annealing at (**a**) room temperature (RT), (**b**) 150 °C, (**c**) 250 °C, and (**d**) 350 °C.

**Figure 5 sensors-20-03842-f005:**
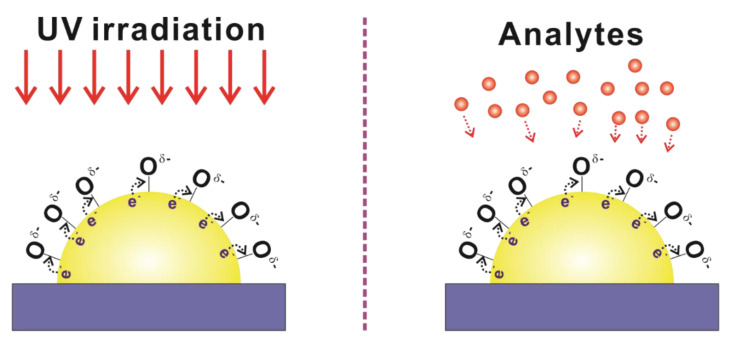
Schematic of the mechanism for the SERS enhancement by the UV irradiation-induced negatively charged adsorbed oxygen species. Under UV irradiation, the excited electrons from the Au bulk transfer into adsorbed oxygen ions, thus increasing the analyte–surface affinity.

**Table 1 sensors-20-03842-t001:** SERS enhancement with UV treatment $\left(I_{UV}-I_{0}\right)/I_{0}\;(\%)$ .

	RT	150 °C	250 °C	350 °C
Au3.4	7.76%	8.95%	3.54%	19.28%
Au4.5	4.68%	14.14%	7.72%	68.75%
Au6.4	46.44%	40.61%	1.3%	7.28%

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
