# Peer review of "UV Irradiation-Induced SERS Enhancement in Randomly Distributed Au Nanostructures"

_sensors, 2020, doi:10.3390/s20143842_

Round 1

Reviewer 1 Report

The work should be completed in experimental sense to avoid misleading conclusions on the interesting UV-irradiation effect in SERS.

Specifically, the authors should make chemically-sensitive XPS or RS analysis of the nanoparticle-based substrate prior and after UV-irradiation. I strongly suspect long-term UV cleaning (removal of adsorbed or physi-sorbed hydrocarbons, nitrogen and oxygen molecules) of the substrate prior SERS measurements, facilitating adsorbance of rhodamine  molecules (see, e.g., surface cleaness effect in SEIRA, Surface-enhanced IR-absorption microscopy of Staphylococcus aureus bacteria on bactericidal nanostructured Si surfaces, Molecules 24 (24), 4488 (2019). On opposite, UV charging is short-term (< 1 ns) relaxation effect, comparing to cleaning. Again, charge transfer implies explicit dipole moment, which is not contributing to SERS, as compared to polarizability.

Next, I would recommend to improve accumulation statisitics by 10 measurements per case (currently - 1 measument).

Finally, I would suggest to combine UV-irradiation flashes with SERS measurements to show onset and decay kinetics of the UV-irradiation effect. 

Overall, the authors observed rather interesting effect, potentially holding a promise of significant advance in the SERS field, but this effect should be studied more accurately and systematically to avoid misleading conclusions.

Reviewer 2 Report

The manuscript seems interesting to me, so we can improve the loads on the surface of a nanostructure. however I would like to verify the following:

  1. You will have a Raman spectrum comparison of rhodamine 6G evaluated with Au nanomaterials with and without UV irradiation. This in order to observe the amplification of the vibrational bands ?. It would be interesting to place the comparative in figure 3a, specifically to observe the behavior of the 611 cm-1 band, in addition to verifying what happens in the other vibrational bands.
  2. In the experimental procedures, in the case of Raman measurements, you must specify in which scanning range you take the spectra of Rhodamine 6G.

And on the other hand,

  1. On line 79 change HORIVA to HORIBA, which is the correct name of the company that manufactures this line of Raman systems.

Round 2

Reviewer 1 Report

After the revision, publish as is.

Reviewer 2 Report

Comments have been addressed, and soon you will receive feedback
from the editor.

Best regards